# 24/7 Therapeutic Drug Monitoring of Beta-Lactam Antibiotics with CLAM-2000

**DOI:** 10.3390/antibiotics12101526

**Published:** 2023-10-10

**Authors:** Tatjana Khromov, Gry Helene Dihazi, Phillipp Brockmeyer, Andreas Fischer, Frank Streit

**Affiliations:** 1Department of Clinical Chemistry, University Medical Center Goettingen, Robert-Koch Str. 40, D-37075 Goettingen, Germany; gryhelene.dihazi@med.uni-goettingen.de (G.H.D.); andreas.fischer@med.uni-goettingen.de (A.F.); frank.streit@med.uni-goettingen.de (F.S.); 2Department of Oral and Maxillofacial Surgery, University Medical Center Goettingen, Robert-Koch Str. 40, D-37075 Goettingen, Germany; ph.brockmeyer@gmail.com

**Keywords:** therapeutic drug monitoring, beta-lactam antibiotics, CLAM-2000

## Abstract

Background: The aim of this study was to evaluate the CLAM-2000 automated preanalytical sample preparation module with integrated liquid chromatography–mass spectrometry/mass spectrometry (LC-MS/MS) as a method for 24/7 therapeutic drug monitoring (TDM) of beta-lactam antibiotics in routine clinical diagnostics. Methods: Method validation was performed using quality control samples. Method comparison was performed with routine samples from patients treated with beta-lactam antibiotics. Results: The determination of piperacillin, meropenem, ceftazidime, flucloxacillin, and cefotaxime was performed using D5-piperacillin and D6-meropenem as internal standards. The linearity of the method was within the therapeutic range of beta-lactam antibiotics. The imprecision and accuracy data obtained from quality control samples were within 15%, and the imprecision of patient samples on the instrument was less than the 5% coefficient of variation (CV). Internal standards stored in the instrument at 9 °C for at least one week were stable, which facilitated reagent use and storage. Conclusion: The CLAM-2000 (Shimadzu, Kyoto, Japan) provides reproducible results as an established routine instrument and is a useful tool for 24/7 TDM of beta-lactam antibiotics in routine clinical diagnostics.

## 1. Introduction

The primary benefit of therapeutic drug monitoring (TDM) is the prevention of toxicity and treatment failure. Targeted, optimized antibiosis can also help prevent bacterial resistance [1,2]. Studies evaluating the clinical outcomes of patients on TDM-guided antibiotic treatment are limited [3], and clear benefits have only been demonstrated for aminoglycosides [3,4,5] and glycopeptides [6]. TDM has gained importance, especially in beta-lactam therapy, with respect to individualized dose adjustment to improve clinical outcomes in critically ill patients with severe sepsis [7,8,9]. An association between beta-lactam antibiotic exposure and clinical outcomes in critically ill patients has been described [10]. Antibiotic dose recommendations and bacterial susceptibility are based primarily on the pharmacokinetics of a “normal patient” [11]. Beta-lactam therapy under standard dosing regimens cannot maintain adequate plasma concentrations in critically ill patients [3,12,13,14,15,16]. Severe sepsis and septic shock can cause significant changes in antibiotic pharmacokinetics. This can lead to disturbances in drug absorption, distribution, metabolism, and excretion, making it difficult to predict drug concentrations [11,17,18,19]. This is true not only for beta-lactam antibiotics [15,16] but also for aminoglycosides [20], glycopeptides [21], fluoroquinolones [22], and oxazolidinones [23]. Drug levels become very erratic, not only in sepsis cases but also in patients with acute renal failure or obese patients [17,18,24]. Changes in the volume of distribution and drug clearance may result in subtherapeutic or toxic antibiotic exposure when standard doses are used [2].

Because the effect of hydrophilic beta-lactam antibiotics is time-dependent, it has been recommended that a 4–10-fold minimum inhibitory concentration (MIC) be maintained for 60–100% of the time during the dosing interval to achieve anti-infective efficacy [3,18,25]. According to the bactericidal pathogen, the beta-lactam concentration should exceed the MIC for the duration of the application (% fT > MIC) [11]. Dosing strategies for beta-lactam antibiotics are a current topic of discussion, especially in critically ill patients. Exact dosing information is difficult to obtain due to the diversity of pathogens and individual patient conditions, such as renal function. However, concentrations of 8–16 mg/L are considered appropriate therapeutic doses, while concentrations of 16–24 mg/L are reported as moderately high, and concentrations >24 mg/L are reported as potentially harmful [26,27].

Continuous beta-lactam application without TDM control may result in a sustained concentration below the pharmacokinetic/pharmacodynamic (PK/PD) target, especially in critically ill patients [11]. Therefore, TDM of beta-lactam antibiotic concentrations and subsequent dose adjustments in these patients can provide maximized therapeutic efficacy with a minimized risk of toxicity associated with renal and/or hepatic impairment [28,29]. The most reliable and widely used TDM method is high-performance liquid chromatography, also combined with fast tandem mass spectrometry (HPLC-MS/MS). Due to the lack of automation, these LC-MS/MS methods are not commonly used in 24/7 emergency laboratories.

In the current investigation, we report an easy-to-use, clinically applicable, routine beta-lactam TDM method using a fully automated sample preparation module connected to an LC-MS/MS system that can be used in a 24/7 environment. The system consists of a Clinical Laboratory Automated Sample Preparation Module (CLAM-2000) connected to an LC-MS/MS system (LC-MS-8050) that bridges the gap between sample preparation and detection. The CLAM-2000, equipped with a user-friendly interface, provides an easy way to determine drug levels around the clock through automated sample preparation and LC-MS analysis. Piperacillin, meropenem, ceftazidime, flucloxacillin, and cefotaxime were used to demonstrate the agreement (sensitivity, specificity, stability, etc.) between our established manual sample preparation followed by LC-MS analysis and the automated CLAM-2000 system. The repeated measurements of the controls within 24 h demonstrated the robustness and routine capability of this instrument. Furthermore, the method comparison showed a high correlation between the samples measured by the CLAM-2000 and our routine method.

## 2. Results

To validate our method, patient samples containing beta-lactam antibiotics (meropenem, flucloxacillin, cefotaxime, ceftazidime, and piperacillin) were measured using the CLAM-2000 LC-MS-8050 automated system and then compared with the evaluated LC-MS/MS method in clinical routine.

### 2.1. Validation of HPLC Assays (Chromatography)

The total run time for all analytes in human plasma was 3.5 min. The retention time was 1.76 min for piperacillin and D5-piperacillin, 0.85 min for meropenem and D6-meropenem, 1.51 min for cefotaxime, 0.68 min for ceftazidime, 1.8 min for flucloxacillin, and 1.79 min for ethylmercaptopurine (EMP; Figure 1). The analytes were detected at the masses described in Section 4.

### 2.2. Linearity

With the CLAM-2000 method, linearity was observed for the antibiotics flucloxacillin, cefotaxime, ceftazidime, and piperacillin over a working range of 0.5–156 mg/L (lower limit of quantitation; LLOQ 0.5 mg/L); for meropenem, the working range was 0.5–62.5 mg/L. The calibration regression curves (y) and the correlation coefficient (R^2^) for the beta-lactam antibiotics are listed in Table 1. The method was linear over the entire concentration range (R^2^ > 0.99).

### 2.3. Accuracy and Imprecision

Accuracy and imprecision were evaluated using three or four different concentrations (10, 25, 50, 150 mg/L), including or excluding the LLOQ, as indicated. Within-run (Table 2) and between-run (Table 3) accuracy and imprecision were evaluated. The within-run imprecision of the instrument was less than 10% for flucloxacillin, meropenem, cefotaxime, and piperacillin. Ceftazidime imprecision was found to be within 15%, while between-run imprecision was less than 10% for all specimens. Accuracy and imprecision were also tested by adding defined concentrations of stock solution to patient samples in equal volumes before and after sample preparation. Patient samples were spiked before and after protein precipitation (supernatant spike) using 0.9% sodium chloride for volume normalization. The extraction efficiency and CV of the analyte area ratio were calculated as 99.45% and 3.3% for flucloxacilline, 91.8% and 7.9% for meropeneme, 97.3% and 7.1% for piperacilline, 87.2% and 9.7% for cefotaxime, and 95.9% and 8.1% for ceftazidime.

### 2.4. Stability Studies

The stability of internal standards, calibrators, quality controls, and patient samples stored in the instrument reagent trays at 9 °C for 24 h was evaluated. The repeatability of the quality controls was less than 10% (Table 4). The repeatability of unstabilized patient samples was less than 5% within 12 h. The stability of the internal standards within one week, as determined by their area under the curve (AUC), was within 10% CV. The overall precision of the patient samples and the quality control samples of beta-lactam antibiotics was less than 10%, meeting the criteria of the European Medicines Agency (EMA) [30].

After six measurements of deuterated internal standards, the imprecision was 12% CV for D6-meropenem and 8.8% CV for D5-piperacillin, demonstrating the robustness and routine capability of this instrument.

### 2.5. Method Correlation

Routine samples from patients treated with piperacillin (*n* = 36), meropenem (*n* = 20), flucloxacillin (*n* = 30), ceftazidime (*n* = 36), or cefotaxime (*n* = 19) were used to compare the CLAM-2000-LC-MS/MS with our routine method (Table 5). The data were analyzed using Passing–Bablok regression. The method comparison showed a good correlation between the CLAM-2000 procedure and our established routine LC-MS/MS method.

### 2.6. Selectivity and Specificity Studies

In addition to observing the multiple reaction monitoring (MRM) transitions of the quantifier and qualifier, we exploited the good selectivity of the LC-MS system with upstream U(H)PLC via chromatographic separation but observed no interference in our measurements.

### 2.7. Robustness

During the three months of instrument use, only reagents and solvent filters were refilled. No mass spectrometer maintenance (ion source or ion guide cleaning) was required during this period.

## 3. Discussion

In the present study, we have established a rapid and robust 24/7 routine method for the rapid analysis of five commonly used beta-lactam antibiotics for individual dose optimization. Standard TDM involves extensive manual pre-analytical steps, including ancillary equipment such as a TECAN liquid handling platform, microtiter plate shaker, and centrifuge for sample preparation before the final LC-MS analysis can be performed. The current CLAM-2000 system, a single instrument coupled to an LC-MS/MS system, enables an innovative approach to TDM in clinical routine, with the advantage of reducing the need for skilled personnel.

Traditional methods for determining drug concentrations are based on automated immunoassays [31] or liquid chromatography. Immunoassays can be affected by matrix interferences and lack of specificity [31]. LC-MS/MS methods have become the gold standard for beta-lactam TDM in clinical routine due to their specificity, precision, and sensitivity [32,33,34,35,36,37,38] and are therefore associated with a high level of expertise and prolonged training [39]. TDM has traditionally been used to minimize the toxic side effects of drugs with narrow therapeutic windows or complex pharmacokinetics [3]. As a result, the methods were well established for evaluating the clinical outcome of aminoglycoside and glycopeptide antibiotic therapy [6]. Many recent studies describe TDM methods using UPLC-MS/MS and HPLC systems for beta-lactam antibiotic dose optimization to improve clinical outcomes in critically ill patients with severe infections [13,38,40,41]. LC-MS/MS instruments are sophisticated and require highly trained personnel to perform preanalytical sample preparation. Therefore, they are not currently suitable for 24/7 emergency use in a clinical laboratory. The easy-to-use automated sample preparation module (CLAM-2000) connects to an LC-MS/MS system with a user-friendly interface. It is similar to the software modules of common clinical chemistry analyzers.

In the first part of the validation during the 24/7 routine, we focused on regularly used beta-lactam antibiotics: three penicillins (flucloxacillin, cefotaxime, piperacillin), one cephalosporin (ceftazidime), and one carbapenem (meropenem). Our data demonstrated the high stability of the internal standards, calibrators, and control material over 24 h on the instrument, as well as the robustness of the new instrument without maintenance for at least three months. The stability of the internal standards was demonstrated at 9 °C for up to 24 h over 10 days on CLAM-2000 reagent trays. The MOPS solution (3-(N-morpholino)propanesulfonic acid), known to stabilize imipenem in plasma samples [42], was used to stabilize all three internal standards (EMP, D6-meropenem, D5-piperacillin). Our observation was that the addition of MOPS to the standards proved to be a critical factor in the overall stability.

Previously, the preanalytical stability of beta-lactams in whole-blood, plasma, and refrigerated and frozen samples has been demonstrated in various studies [43,44,45,46]. Meropenem and piperacillin in whole-blood and plasma samples were stable for 8 h and 48 h at 4 °C, respectively [12]. In addition, both antibiotics were stable for 6 h after storage at room temperature for 4 h followed by storage at 4 °C after gel separation [12]. Similarly, ceftazidime and flucloxacillin were stable in plasma for 12 h, both at 4 °C on the autosampler and at room temperature (RT) [12,13,47]. Meropenem was stable for 8 h at 4 °C and 3–6 h at RT in both whole blood and plasma, while piperacillin was stable for 48 h in plasma without gel separation [40]. To ensure sample stability, we recommend a transport time of less than 4 h at RT or less than 24 h at 4 °C. The analytes were stable for several days at −20 °C.

The between-run imprecision of the internal standards D6-meropenem and D5-piperacillin was less than 15% CV, demonstrating the robustness and routine capability of the CLAM-2000-LC-MS/MS system. The imprecision of the 10 quality control samples at each concentration level, including the LLOQ, evaluated on the same day (within-run) and over the next 10 days (between-run), was found to be less than 15%, meeting the criteria of the EMA (guideline version EMEA/CHMP/EWP/192217/2009 Rev. 1 Corr. 2) and the U.S. Department of Health and Human Services Food and Drug Administration (FDA, FDA-2013-D-1020). These results demonstrate the acceptable precision and authenticity of the CLAM-2000 TDM method.

## 4. Materials and Methods

### 4.1. Chemicals and Reagents

Calibrators (10 and 25 mg/mL) and quality controls (10, 25, 50, and 150 mg/mL) were prepared by adding high-purity powder of beta-lactams at specific concentrations to drug-free serum (Recipe, Munich, Germany) dissolved in 99.5% MOPS (Sigma-Aldrich, Schnelldorf, Germany). Piperacillin, flucloxacillin, and cefotaxime were obtained from Sigma-Aldrich; meropenem from the European Pharmacopoeia (Strasbourg, France); and ceftazidime from the U.S. Pharmacopeia (Rockville, MD, USA). D5-piperacillin and D6-meropenem were purchased from AlsaChim (Duisburg, Germany), and EMP was purchased from Sigma-Aldrich (Schnelldorf, Germany). Separate stock solutions were prepared for calibrators and QC samples. All calibrators and QC samples were aliquoted and stored at −80 °C after preparation. Stock solutions of internal standards (EMP, D6-meropenem, and D5-piperacillin) were prepared at 1 g/L in 50% methanol (ChemSolute, Th. Geyer, Renningen, Germany) and stored at −80 °C. Acetonitrile (ChemSolute, Th. Geyer, Renningen, Germany), ammonium acetate (Sigma-Aldrich, Schnelldorf, Germany), and deionized water were used as mobile phases.

### 4.2. Matrix Effects

To investigate and exclude matrix effects, we performed different types of experiments: the response of the isotope-labeled internal standard was tested with different sample matrices and found to be less than 15% overall. In addition, three different patient samples and MOPS buffer were spiked 3:1, 1:1, and 1:3 with a quality control. The CV% of the measurement results was at least <10% for all matrices and different concentrations. The overall precision was less than 10%. Specifically, the CV% of the analyte area from all the above spiked matrices corrected for dilution was 4.6% for flucloxacilline, 6.8% for meropeneme, 6.6% for piperacilline, 7.9% for cefotaxime, and 9.6% for ceftazidime.

### 4.3. Analysis of Beta-Lactam Antibiotics in a 24/7 Environment (CLAM-2000)

The validation of beta-lactam antibiotics for routine 24/7 clinical diagnosis was performed using the CLAM-2000 (Shimadzu Corporation, Kyoto, Japan) connected to a UHPLC-MS/MS system (NexeraX2-LC-MS-8050, Shimadzu, Kyoto, Japan). CLAM-2000 (in combination with LabSolutions software, version 5.1.09) was used for instrument control, data acquisition, and processing.

#### 4.3.1. Sample Preparation

Blood collected in heparinized tubes (S-Monovette Lithium-Heparin Gel, Sarstedt, Nümbrecht, Germany) from clinical routine was used for method comparison. The samples were centrifuged (15 min, 3700 rpm) to obtain plasma for further procedures. Calibrators and quality control samples were prepared independently and inoculated with the antibiotic stock solution.

Automated sample preparation of calibrators, QCs, and patient samples was performed using the CLAM-2000 module (Shimadzu, Kyoto). The stabilized internal standard, precipitation solution, calibrators, and QCs were loaded into the CLAM-2000 trays daily. Patient plasma samples were placed on the instrument throughout the day. Protein precipitation of these samples was performed by adding 10 µL calibrator/QC/patient sample to 115 µL precipitation reagent containing MeOH, H_2_O, deuterated internal standard or EMP, and acetonitrile. After shaking the mixture (30 s) and filtration (90 s) on the CLAM to separate the precipitated proteins from the analytes, the filtrate was diluted with 150 µL H_2_O. Then, 1 µL was injected into the chromatographic column.

#### 4.3.2. Separation (Chromatographic Conditions)

For chromatographic separation, a sharp linear gradient was run on a BEH shield C18 column (2.1 × 100 mm, 1.7 µm) (Waters Corporation, Milford, MA, USA) at 45 °C. Mobile phases A (4 mM ammonium acetate and 5% acetonitrile) and B (100% acetonitrile) were pumped at a flow rate of 0.500 mL/min in gradient mode. After injection of 1 µL filtrate, the gradient started at 5% B and was increased linearly to 45% B from 0.42 to 0.92 min, then to 65% B from 0.92 to 1.42 min, and held at 65% until 1.70 min. The gradient was then set to 95% B for a further 0.8 min. The column was re-equilibrated to initial conditions for 0.7 min. The total run time was 3.2 min.

#### 4.3.3. Mass Spectrometry

All experiments were performed on a triple quadrupole mass spectrometer equipped with an electrospray ionization (ESI) operating in positive ion mode. The analytes were fragmented using argon 5.0 as the collision gas, and the optimized MRM transitions for the given analytes were monitored. All calculations were based on the peak area ratios of the given beta-lactam antibiotic and internal standards. The following MRM transitions were used for quantification: *m*/*z* 384.1/141.0 for meropenem, *m*/*z* 547.0/467.9 for ceftazidime, *m*/*z* 455.9/324.1 for cefotaxime, *m*/*z* 453. 9/159.9 for flucloxacillin, *m*/*z* 518.0/143.1 for piperacillin, *m*/*z* 181.0/125.0 for EMP, *m*/*z* 523.0/148.0 for D5-piperacillin, and *m*/*z* 390.0/147.0 for D6-meropenem.

The nebulizing gas was set at 5 L/min, the heating flow gas at 10 L/min, and the drying gas at 10 L/min. The interface temperature was 300 °C, the DL temperature was 200 °C, and the heating block temperature was 400 °C.

#### 4.3.4. Validation Assay

Assay validation was performed according to the EMA guideline on bioanalytical method validation [30] or the FDA industry guidance [48].

#### 4.3.5. Accuracy and Precision

At least 10 determinations per concentration of quality control samples at four different concentrations, including the LLOQ, were analyzed in a batch (within-run) and over several days (between-run). Precision did not exceed 15% of the CV, except for the LLOQ (20%). Accuracy was calculated as the deviation of the mean from the true value and was within 15% of the true value, except for the LLOQ (20%).

#### 4.3.6. Linearity

The linearity of the method was evaluated in human plasma using five spiked samples of meropenem, flucloxacillin, cefotaxime, ceftazidime, and piperacillin at a minimum of 10 concentration levels. For each analyte, linear regression was performed. The LLOQ, determined as the lowest concentration, was accurately and precisely quantified.

#### 4.3.7. Stability and Robustness

The stability of the internal standards in the aqueous solution was evaluated by repeated measurements (AUC) over 10 days. Using these stabilized internal standards, the MOPS-stabilized quality controls, stored in the instrument at 9 °C for 24 h, were measured against freshly thawed calibrators. Acceptance criteria for inaccuracy and imprecision were considered.

### 4.4. Comparison with Routine Method

The established method with the CLAM-2000 was compared with our evaluated LC-MS/MS method in clinical routine for the quantitative analysis of beta-lactam antibiotics in heparin plasma.

Routine use was made of the UPLC Classic (Waters Corporation, Milford, MA, USA) chromatography system coupled to a Xevo TQS Triple Quad mass spectrometer (Waters Corporation, Milford, MA, USA) equipped with an ESI interface and running Mass Lynx 4.1 software. A Tecan Genesis RSP 150 automated liquid handling system (Tecan Group Ltd., Männedorf, Switzerland) was used for sample preparation.

Protein precipitation of the given beta-lactams from human plasma samples was performed on the Tecan Genesis RSP 150 as follows: a 50 µL patient sample was treated with a 200 µL precipitation reagent containing 50 mL MeOH, 50 µL internal standard (EMP, 1 g/L), and phosphoric acid (6%) and mixed vigorously. After centrifugation, the supernatant was diluted 2:1 with water and injected into the LC-MS/MS system (0.1 µL).

## 5. Conclusions

Our study shows a high positive correlation between the CLAM-2000 TDM procedure and our established routine LC-MS method. This highlights the CLAM system as a next-generation instrument that offers the benefits of automation for easy pre-analytical sample preparation and, thus, via the optimized method for reagent stability, 24/7 measurement of beta-lactam antibiotics.

## Figures and Tables

**Figure 1 antibiotics-12-01526-f001:**
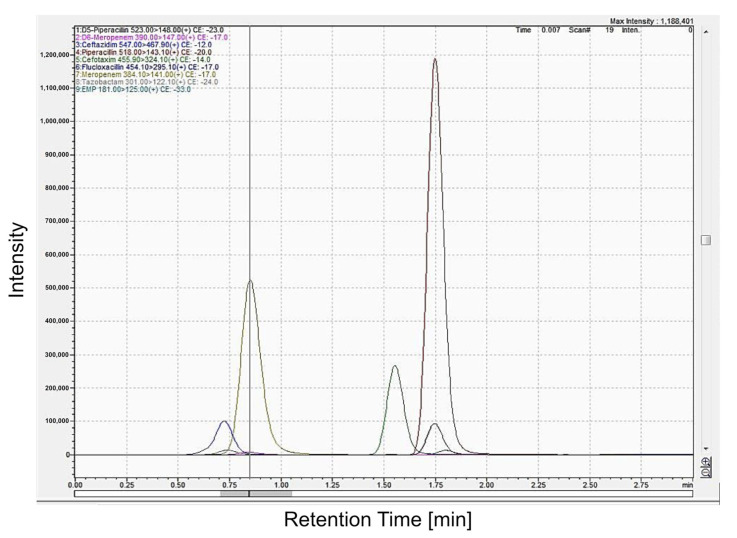
UHPLC-MS/MS chromatograms of the five beta-lactam calibrators at 25 mg/L.

**Table 1 antibiotics-12-01526-t001:** Linearity of beta-lactam antibiotics.

Analyte	Y	R^2^	Range (mg/L)
Ceftazidime	1.0086x − 0.7149	0.9992	0.5–156
Cefotaxime	0.9888x − 0.4574	0.9961	0.5–156
Flucloxacillin	1.0374x + 0.7133	0.9987	0.5–156
Meropenem	1.0033x + 0.2383	0.9994	0.5–62.5
Piperacillin	0.9758x + 2.0205	0.9932	0.5–156

**Table 2 antibiotics-12-01526-t002:** Within-run analytical accuracy and imprecision of beta-lactam antibiotics (*n* = 10).

Sample	Concentration (mg/L)	Nominal Value (mg/L)	Concentration Found (mg/L) (Mean ± SD)	Imprecision CV (%)	Bias d (%)
Ceftazidime					
LLOQ	0.35	0.35	0.3 ± 0.04	11.7	3.9
QC 1	10	10	10.5 ± 1.0	9.5	−4.5
QC 2	25	25	25.5 ± 2.1	8.03	−1.9
QC 3	50	51	49 ± 4	8.2	3.9
QC 4	150	191	174.6 ± 17.9	10.3	−8.6
Cefotaxime					
LLOQ	0.5	0.5	0.5 ± 0.03	6.1	3.5
QC 1	10	8.7	9 ± 0.3	3.1	−3.2
QC 2	25	20	20.3 ± 0.8	3.7	−1.6
QC 3	50	46	47.4 ± 2.2	4.6	−3.1
QC 4	150	163	161.4 ± 9.4	5.8	−1
Flucloxacillin					
LLOQ	0.5	0.5	0.5 ± 0.04	7.9	5.8
QC 1	10	9	8.9 ± 0.5	5.2	1.7
QC 2	25	25	24.3 ± 1.6	6.6	2.9
QC 3	50	46	46.4 ± 2.4	5.2	−0.9
QC 4	150	--	--	--	--
Meropenem					
LLOQ	0.5	0.5	0.5 ± 0.02	4.2	0.9
QC 1	10	10	10.2 ± 0.5	4.6	−2
QC 2	25	25	24.1 ± 1.1	4.6	3.5
QC 3	50	50	51.9 ± 2.5	4.9	−3.8
QC 4	150	160	162.2 ± 3.9	2.4	1.4
Piperacillin					
LLOQ	0.5	0.5	0.5 ± 0.02	3.8	4.8
QC 1	10	12	11.5 ± 0.5	4.6	4
QC 2	25	28	28 ± 0.7	2.5	−0.1
QC 3	50	55	56± 1.6	2.8	−1.8
QC 4	150	150	142.8 ± 2.3	1.6	−4.8

**Table 3 antibiotics-12-01526-t003:** Between-run analytical precision and imprecision of beta-lactam antibiotics (*n* = 10).

Sample	Concentration (mg/L)	Nominal Value (mg/L)	Concentration Found (mg/L) (Mean ± SD)	Imprecision CV (%)	Bias d (%)
Ceftazidime					
QC 1	10	10	10.5 ± 0.8	7.2	4.6
QC 2	25	25	26.7 ± 1.7	6.3	6.9
QC 3	50	51	49.9 ± 4.8	9.7	2.2
QC 4	150	191	179.2 ± 13.1	7.3	6.2
Cefotaxime					
QC 1	10	8.7	8.8 ± 0.1	1.6	0.6
QC 2	25	20	19.7 ± 0.7	3.8	1.5
QC 3	50	46	46.7 ± 0.9	2	1.4
QC 4	150	163	164.8 ± 6.2	3.8	1.1
Flucloxacillin					
QC 1	10	10	9 ± 0.4	4.04	1
QC 2	25	25	24.9 ± 1.5	6.03	0.5
QC 3	50	46	47.2 ± 2.5	5.3	2.6
QC 4	150	--	--	--	--
Meropenem					
QC 1	10	10	10.1 ± 0.3	2.8	0.5
QC 2	25	23	22.9 ± 0.6	2.5	0.3
QC 3	50	50	50.6 ± 2	3.9	1.2
QC 4	150	160	169.2 ± 5.1	3	5.8
Piperacillin					
QC 1	10	10	10.5 ± 0.3	3.2	5.2
QC 2	25	28	25.7 ± 1.1	4.2	8.1
QC 3	50	55	51 ± 1.1	2.2	5
QC 4	150	180	179.4 ± 8.8	7.3	0.4

**Table 4 antibiotics-12-01526-t004:** Stability of controls stored on device for 24 h (*n* = 10).

Analyte	c (mg/L)	CV (%)	d (%)
Ceftazidime	10	9.5	4.5
	25	8.0	1.9
	50	8.2	3.9
Cefotaxime	9	3.1	1.6
	20	3.7	1.6
	46	4.6	3.9
Flucloxacillin	9	5.2	1.7
	25	6.6	2.9
	46	5.2	0.9
Meropenem	10	4.6	2.0
	25	4.6	3.5
	50	4.9	3.8
Piperacillin	12	4.6	4.0
	28	2.5	0.1
	55	2.8	1.8

**Table 5 antibiotics-12-01526-t005:** Method comparison UPLC-MS/MS vs. CLAM-2000 LC-MS-8050 of five beta-lactam antibiotics.

Analyte	*n*	R	Slope (b)	Intercept (a)	Range MSMS	Range CLAM-2000
Ceftazidime	39	0.994	0.993	−0.605	3.1–187	2.9–196
Cefotaxime	19	0.994	0.959	0.956	4.6–54	4.6–52
Flucloxacillin	20	0.981	0.924	−0.269	9.8–47.9	8.3–49.6
Meropenem	30	0.981	1.058	−9.41	5.4–155	4.8–190
Piperacillin	36	0.983	1.076	−5.03	29.1–343	22–350.9

## Data Availability

All data are available from the corresponding author.

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
