# Peer review of "24/7 Therapeutic Drug Monitoring of Beta-Lactam Antibiotics with CLAM-2000"

_antibiotics, 2023, doi:10.3390/antibiotics12101526_

Round 1

Reviewer 1 Report

The manuscript describes evaluation of the CLAM-2000 automated preanalytical sample preparation module with integrated liquid chromatography mass spectrometry/mass 13 spectrometry (LC-MS/MS) as a method for measuring beta lactam antibiotics in real time, i.e., when it is needed in clinical settings of complex ICU patients who need frequent adjustments of antibiotic dosage regimens. The combination of automated plasma sample processing and preparation of LC-MS/MS enables easy use of the method in clinical settings, which is for now very hard to achieve with current technology; therefore this combinations increases clinical utility greatly.

The manuscript is written clearly, in good language and style. Methodology of the measurement method evaluation is appropriate, since it was compared with gold standard of measurement, and sufficient replication of measurements was made. Precision and accuracy were calculated, as well as correlation of measurements with gold standard. The results are presented clearly, in appropriate tables. References used are relevant. The content of the manuscript has both novelty and clinical significance. I recommend acceptance of the manuscript in its present form.

Author Response

Authors respond: We would like to thank reviewer 1 for the critical evaluation of the manuscript and the praise for our work.

Reviewer 2 Report

The authors developed a new automated method that would enable 24/7 therapeutic monitoring of beta-lactam antibiotics (important subject in TDM field). They provide data on linearity, accuracy and imprecision, stability studies, comparison of the new automated method to standard UPLC-MS/MS method and selectivity and specificity studies, all the requirements for developing a new method.

Major concerns:

1. QCs were prepared in drug-free serum, while plasma from patients was used for method comparison. The authors should show that there is no matrix effect on the final result;

2. Accuracy and imprecision should also be performed in spiked plasma samples (from patients), as the analytes may behave differently in extracted plasma samples because of different interferences;

3. The authors should also test for extraction efficiency by spiking plasma before extraction and adding the same amount of analyte to non-spiked samples after extraction (prior to MS analyses).

Minor concerns:

1. In Figure 1. axes should be named (although for researchers from the field it is clear that this is RT and abundance);

2. In the introduction part the authors should describe which are the clinically relevant drug concentrations in patients.

Author Response

Major concerns:

1.) QCs were prepared in drug-free serum, while plasma from patients was used for method comparison. The authors should show that there is no matrix effect on the final result.

Authors respond: We would like to thank reviewer 2 for critically evaluating our manuscript and providing useful comments to improve the quality of the paper. To investigate and exclude matrix effects, we performed additional different types of experiments: The response of the isotope-labelled internal standard was tested with different sample matrices and found to be less than 15% overall. In addition, three different patient samples and MOPS buffer were spiked 3:1, 1:1 and 1:3 with a quality control. The CV% of the measurement results was at least <10% for all matrices and different concentrations. The overall precision was less than 10%. Specifically, the CV% of the analyte area from all the above spiked matrices corrected for dilution were as follows 4.6% for flucloxacilline, 6.8% for meropeneme, 6.6% for piperacilline, 7.9% for cefotaxime and 9.6% for ceftazidime. We have added the corresponding information’s to the Methods section of the manuscript. We have also uploaded the raw data sets of the control experiments for the reviewer.

2.) Accuracy and imprecision should also be performed in spiked plasma samples (from patients), as the analytes may behave differently in extracted plasma samples because of different interferences;

Authors respond: Accuracy and imprecision were also tested by adding defined concentrations of stock solution to patient samples in equal volumes before and after sample preparation. Patient samples were spiked before and after protein precipitation (supernatant spike) using 0.9% sodium chloride for volume normalisation.

3.) The authors should also test for extraction efficiency by spiking plasma before extraction and adding the same amount of analyte to non-spiked samples after extraction (prior to MS analyses).

Authors respond: Extraction efficiency and CV of analyte area ratio were calculated. 99.45% and 3.3% for flucloxacilline, 91.8% and 7.9% for meropeneme, 97.3% and 7.1% for piperacilline, 87.2% and 9.7% for cefotaxime and 95.9% and 8.1% for ceftazidime.

Minor concerns:

1.) In Figure 1. axes should be named (although for researchers from the field it is clear that this is RT and abundance);

Authors respond: We are grateful for this useful advice, and have revised Figure 1 accordingly and included it in the manuscript.

2.) In the introduction part the authors should describe which are the clinically relevant drug concentrations in patients. 

Authors respond: Dosing strategies for beta-lactam antibiotics are a current topic of discussion, especially in critically ill patients. Exact dosages cannot be given due to the diversity of pathogens and individual patient conditions. However, concentrations of 8-16 mg/L are reported in the literature as adequate therapeutic doses, concentrations of 16-24 mg/L as moderately high concentrations, and concentrations >24 mg/L as potentially harmful concentrations. We have included a reference to this in the manuscript:

Chiriac, U., Richter, D., Frey, O. R., Röhr, A. C., Helbig, S., Hagel, S., ... & Brinkmann, A. (2023). Software-and TDM-Guided Dosing of Meropenem Promises High Rates of Target Attainment in Critically Ill Patients. Antibiotics, 12(7), 1112.

Reviewer 3 Report

The topic of the manuscript is interesting. However, the manuscript need various amendments before it can be accepted/

1) Table 1, why the nominal value is 035?

2) THe matrix effect is not analyzed.

3) The incurred sample reanalysis is missing. 

4) How good is the analytes in plasma?

Author Response

1.) Table 1, why the nominal value is 035? 

Authors respond: We would like to thank reviewer 3 for reviewing our manuscript and appreciate his advice. We assume that you mean Table 2 and not Table 1. You are correct, unfortunately we made a transcription error when creating Table 2. We have corrected the error in the manuscript.

2.) The matrix effect is not analyzed.

Authors respond: Thank you for this useful advice. To investigate and exclude matrix effects, we additionally performed different types of experiments: The response of the isotope-labelled internal standard was tested with different sample matrices and found to be less than 15% overall. In addition, three different patient samples and MOPS buffer were spiked 3:1, 1:1 and 1:3 with a quality control. The CV% of the measurement results was at least <10% for all matrices and different concentrations. The overall precision was less than 10%. Specifically, the CV% of the analyte area from all the above spiked matrices corrected for dilution was 4.6% for flucloxacilline, 6.8% for meropeneme, 6.6% for piperacilline, 7.9% for cefotaxime and 9.6% for ceftazidime. The following information has been added to the manuscript.

3.) The incurred sample reanalysis is missing.

Authors respond: Parts of method development was stability experiments with qc as well as patient samples. Calibrators and quality controls were prepared every 6 month with a quarternarly shift to exclude drift in calibration. Calibrator are compared at each lot change. CV% and accuracy were less than 15%. External quality controls have been exchanged between other labs. Commercially available control sample have been used until on market.

4.) How good is the analytes in plasma?

Authors respond: Thank you for this useful suggestion. Accuracy and imprecision were also tested by adding defined concentrations of stock solution to patient samples in equal volumes before and after sample preparation. Patient samples were spiked before and after protein precipitation (supernatant spike) with 0.9% sodium chloride for volume normalisation. A reference has been added to the manuscript.

Reviewer 4 Report

The authors have carried out interesting investigation of TDM of beta-lactam antibiotic levels during routine clinical diagnosis. Automated CLAM-2000 equipment was coupled with LC-MS/MS for method development and quantitation of antibiotics.  The manuscript needs minor changes to formatting and responses to following questions:

1.       Please convert mg/l to mg/mL or µg/mL throughout the manuscript.

2.       On lines 182-189, while stating criteria by EMA and FDA could you please incorporate corresponding regulatory document references.

3.       What are pros and cons of using CLAM-2000 (Shimadzu Corporation, Kyoto, Japan) in the 24/7 TDM compared to dried blood spot quantitation.

Thank you.

Author Response

1.) Please convert mg/l to mg/mL or µg/mL throughout the manuscript.

Authors respond: We would like to thank reviewer 3 for reviewing the manuscript and providing useful suggestions. Since our method validation was performed in mg/L and the reference ranges are given in this unit and are used in clinical routine, we do not consider it useful to change the units. Furthermore, comparable measurements are performed in this unit in all laboratories known to us so far. Some references are given below:

Chiriac, U., Richter, D., Frey, O. R., Röhr, A. C., Helbig, S., Hagel, S., ... & Brinkmann, A. (2023). Software-and TDM-Guided Dosing of Meropenem Promises High Rates of Target Attainment in Critically Ill Patients. Antibiotics, 12(7), 1112.

Scharf, C., Paal, M., Schroeder, I., Vogeser, M., Draenert, R., Irlbeck, M., ... & Liebchen, U. (2020). Therapeutic drug monitoring of meropenem and piperacillin in critical illness—experience and recommendations from one year in routine clinical practice. Antibiotics, 9(3), 131.

2.) On lines 182-189, while stating criteria by EMA and FDA could you please incorporate corresponding regulatory document references.

Authors respond: Thank you for this comment. We have referred to the relevant legal documents in the manuscript.

3.) What are pros and cons of using CLAM-2000 (Shimadzu Corporation, Kyoto, Japan) in the 24/7 TDM compared to dried blood spot quantitation.

Authors respond: Thank you for this helpful comment. The quantification of dried blood spots is not established in routine clinical diagnostics outside of neonatal screening in maximum care hospitals, especially in critically ill patients. Blood plasma is used for antibiotic TDM. CLAM-2000 requires only two steps to start the analysis process. After patient samples and controls have been placed in the system and the probe barcode has been automatically read, the desired application can be started in the software module. As the calibration is stable over one week, no further recalibration is required. With this new technology, the hands-on time required for clinical work can be drastically reduced. Once the analysis is complete, with each run taking no more than five minutes, the results are transmitted via the HL7 interface. Within 16 hours, samples can be placed and measured until QC samples are outside the acceptance criteria. QC samples are measured according to Rilibäk (DOI: 10.3238/arztebl.2019.rili_baek_QS_Labor20222511). CLAM-2000 can be used by untrained personnel not specialized in LCMS instrumentation.

Round 2

Reviewer 2 Report

The authors addressed all the concerns and have significantly improved the strength of their experiments from the V1 of the paper.